# Learning with Invariance via Linear Functionals on Reproducing Kernel Hilbert Space

**Xinhua Zhang**
Machine Learning Research Group
National ICT Australia and ANU
xinhua.zhang@nicta.com.au

**Wee Sun Lee**
Department of Computer Science
National University of Singapore
leews@comp.nus.edu.sg

**Yee Whye Teh**
Department of Statistics
University of Oxford
y.w.teh@stats.ox.ac.uk

## Abstract

Incorporating invariance information is important for many learning problems. To exploit invariances, most existing methods resort to approximations that either lead to expensive optimization problems such as semi-definite programming, or rely on separation oracles to retain tractability. Some methods further limit the space of functions and settle for non-convex models. In this paper, we propose a framework for learning in reproducing kernel Hilbert spaces (RKHS) using local invariances that explicitly characterize the behavior of the target function around data instances. These invariances are *compactly* encoded as linear functionals whose value are penalized by some loss function. Based on a representer theorem that we establish, our formulation can be efficiently optimized via a convex program. For the representer theorem to hold, the linear functionals are required to be bounded in the RKHS, and we show that this is true for a variety of commonly used RKHS and invariances. Experiments on learning with unlabeled data and transform invariances show that the proposed method yields better or similar results compared with the state of the art.

## 1 Introduction

Invariances are among the most useful prior information used in machine learning [1]. In many vision problems such as handwritten digit recognition, detectors are often supposed to be invariant to certain local transformations, such as translation, rotation, and scaling [2, 3]. One way to utilize this invariance is by assuming that the gradient of the function is small along the directions of transformation at each data instance. Another important scenario is semi-supervised learning [4], which relies on reasonable priors over the relationship between the data distribution and the discriminant function [5, 6]. It is commonly assumed that the function does not change much in the proximity of each observed data instance, which reflects the typical clustering structure in the data set: instances from the same class are clustered together and away from those of different classes [7–9]. Another popular assumption is that the function varies smoothly over the graph Laplacian [10–12].

A number of existing works have established a mathematical framework for learning with invariance. Suppose $T(x, \theta)$ transforms a data point $x$ by an operator $T$ with a parameter $\theta$ (*e.g.* $T$ for rotation and $\theta$ for the degree of rotation). Then to incorporate invariance, the target function $f$ is assumed to be (almost) invariant over $T(x, \Theta) := \{T(x, \theta) : \theta \in \Theta\}$, where $\Theta$ controls the locality of invariance. The consistency of this framework was shown by [13] in the context of robust optimization [14]. However, in practice it usually leads to a large or infinite number of constraints, and hence tractable formulations inevitably rely on approximating or restricting the invariance under consideration. Finally, this paradigm gets further complicated when $f$ comes from a rich space of functions, *e.g.* the reproducing kernel Hilbert space (RKHS) induced by universal kernels.

In [15], all perturbations within the ellipsoids around instances are treated as invariances. This led to a second order cone program, which is difficult to solve efficiently. In [16], a discrete set of $\Theta$ that corresponds to feature deletions [17] are considered. The problem is reduced to a quadratic program, but at the cost of blowing up the number of variables which makes scaling to large problems chal-

lenging. A one step approximation of $T(x, \Theta)$ via the even-order Taylor expansions around $\theta = 0$ is used in [18]. This results in a semi-definite programming, which is still hard to solve. A further simplification was introduced by [19], which performed sparse approximations of $T(x, \Theta)$ by finding (via an oracle) the most violating instance under the current solution. Besides yielding a cheap quadratic program at each iteration, it also improved upon the Virtual Support Vector approach in [20], which did not have a clear optimization objective despite a similar motivation of sparse approximation. However, tractable oracles are often unavailable, and so a simpler approximation can be performed by merely enforcing the invariance of $f$ along some given directions, *e.g.* the tangent direction $\frac{\partial}{\partial \theta}|_{\theta=0} T(x, \theta)$. This idea was used in [21] in a nonlinear RKHS, but their direction of perturbation was *not* in the original space but in the RKHS. By contrast, [3] did penalize the gradient of $f$ in the original feature space, but their function space was limited to neural networks and only locally optimal solutions were found.

The goal of our paper, therefore, is to develop a new framework that: (1) allows a variety of invariances to be compactly encoded over a rich family of functions like RKHS, and (2) allows the search of the optimal function to be formulated as a convex program that is efficiently solvable. The key requirement to our approach is that the invariances can be characterized by linear functionals that are bounded (§ 2). Under this assumption, we are able to formulate our model into a standard regularized risk minimization problem, where the objective consists of the sum of loss functions on these linear functionals, the usual loss functions on the labeled training data, and a regularization penalty based on the RKHS norm of the function (§ 3). We give a representer theorem that guarantees that the cost can be minimized by linearly combining a finite number of basis functions[1]. Using convex loss, the resulting optimization problem is a convex program, which can be efficiently solved in a batch or online fashion (§ 5). Note [23] also proposed an operator based model for invariance, but did not derive a representer theorem and did not study the empirical performance.

We also show that a wide range of commonly used invariances can be encoded as bounded linear functionals. This include derivatives, transformation invariances, and local averages in commonly used RKHSs such as those defined by Gaussian and polynomial kernels (§ 4). Experiment show that the use of some of these invariances within our framework yield better or similar results compared to the state of the art.

Finally, we point out that our focus is to find a function in a *given* RKHS which respects the pre-specified invariances. We are *not* constructing kernels that instantiate the invariance, *e.g.* [24, 25].

## 2 Preliminaries

Suppose features of training examples lie in a domain $\mathcal{X}$. A function $k : \mathcal{X} \times \mathcal{X} \to \mathbb{R}$ is called a *positive semi-definite kernel* (or simply kernel) if for all $l \in \mathbb{N}$ and all $x_1, \ldots, x_l \in \mathcal{X}$, the $l \times l$ Gram matrix $K := (k(x_i, x_j))_{ij}$ is symmetric positive semi-definite. Example kernels on $\mathbb{R}^n \times \mathbb{R}^n$ include polynomial kernel of degree $r$, which are defined as $k(\mathbf{x}_1, \mathbf{x}_2) = (\mathbf{x}_1 \cdot \mathbf{x}_2 + 1)^r$,[2] as well as Gaussian kernels, defined as $k(\mathbf{x}_1, \mathbf{x}_2) = \kappa_\sigma(\mathbf{x}_1, \mathbf{x}_2)$ where $\kappa_\sigma(\mathbf{x}_1, \mathbf{x}_2) := \exp(-\|\mathbf{x}_1 - \mathbf{x}_2\|^2 / (2\sigma^2))$. More comprehensive introductions to kernels are available in, *e.g.*, [26–28].

Given $k$, let $\mathcal{H}_0$ be the set of all finite linear combinations of functions in $\{k(x, \cdot) : x \in \mathcal{X}\}$, and endow on $\mathcal{H}_0$ an inner product as $\langle f, g \rangle = \sum_{i=1}^p \sum_{j=1}^q \alpha_i \beta_j k(x_i, y_j)$ where $f(\cdot) = \sum_{i=1}^p \alpha_i k(x_i, \cdot)$ and $g(\cdot) = \sum_{j=1}^q \beta_j k(y_j, \cdot)$. Note $\langle f, g \rangle$ is invariant to the form of expansion of $f$ and $g$ [26, 27]. Using the positive semi-definite properties of $k$, it is easy to show that $\mathcal{H}_0$ is an inner product space, and we call its completion under $\langle \cdot, \cdot \rangle$ as a reproducing kernel Hilbert space (RKHS) $\mathcal{H}$ induced by $k$. For any $f \in \mathcal{H}$, the reproducing property implies $f(x) = \langle f, k(x, \cdot) \rangle$ and we denote $\|f\|^2 := \langle f, f \rangle$.

### 2.1 Operators and Representers

**Definition 1** (Bounded Linear Operator and Functional). *A linear operator $T$ is a mapping from a vector space $\mathcal{V}$ to a vector space $\mathcal{W}$, such that $T(x + y) = Tx + Ty$ and $T(\alpha x) = \alpha \cdot Tx$ for all $x, y \in \mathcal{V}$ and scalar $\alpha \in \mathbb{R}$. $T$ is also called a* functional *if $\mathcal{W} = \mathbb{R}$. In the case that $\mathcal{V}$ and $\mathcal{W}$ are normed, $T$ is called bounded if $c := \sup_{x \in \mathcal{V}, \|x\|_\mathcal{V}=1} \|Tx\|_\mathcal{W}$ is finite, and we call $c$ as the norm of the operator, denoted by $\|T\|$.*

**Example 1.** *Let $\mathcal{H}$ be an RKHS induced by a kernel $k$ defined on $\mathcal{X} \times \mathcal{X}$. Then for any $x \in \mathcal{X}$, the linear functional $T : \mathcal{H} \to \mathbb{R}$ defined as $T(f) := f(x)$ is bounded since $|f(x)| = |\langle f, k(x, \cdot)\rangle| \leq \|k(x, \cdot)\| \cdot \|f\| = k(x, x)^{1/2} \|f\|$ by the Cauchy-Schwarz inequality.*

Boundedness of linear functionals is particularly useful thanks to the Riesz representation theorem which establishes their one-to-one correspondence to $\mathcal{V}$ [29].

**Theorem 1** (Riesz representation Theorem). *Every bounded linear functional $L$ on a Hilbert space $\mathcal{V}$ can be represented in terms of an inner product $L(x) = \langle x, z\rangle$ for all $x \in \mathcal{V}$, where the representer of the functional, $z \in \mathcal{V}$, has norm $\|z\| = \|L\|$ and is uniquely determined by $L$.*

**Example 2.** *Let $\mathcal{H}$ be the RKHS induced by a kernel $k$. For any functional $L$ on $\mathcal{H}$, the representer $z$ can be constructed as $z(x) = \langle z, k(x, \cdot)\rangle = L(k(x, \cdot))$ for all $x \in \mathcal{X}$. By Theorem 1, $z \in \mathcal{H}$.*

Using Riesz's representer theorem, it is not hard to show that for any bounded linear operator $T : \mathcal{V} \to \mathcal{V}$ where $\mathcal{V}$ is Hilbertian, there exists a unique bounded linear operator $T^* : \mathcal{V} \to \mathcal{V}$ such that $\langle Tx, y\rangle = \langle x, T^*y\rangle$ for all $x, y \in \mathcal{V}$. $T^*$ is called the *adjoint operator*. So continuing Example 2:

**Example 3.** *Suppose the functional $L$ has the form $L(f) = T(f)(x_0)$, where $x_0 \in \mathcal{X}$ and $T : \mathcal{H} \to \mathcal{H}$ is a bounded linear operator on $\mathcal{H}$. Then the representer of $L$ is $z = T^*(k(x_0, \cdot))$ because*

$$\forall\, x \in \mathcal{X}, z(x) = L(k(x, \cdot)) = T(k(x, \cdot))(x_0) = \langle T(k(x, \cdot)), k(x_0, \cdot)\rangle = \langle k(x, \cdot), T^*(k(x_0, \cdot))\rangle. \quad (1)$$

Riesz's theorem will be useful for our framework since it allows us to compactly represent functionals related to local invariances as elements of the RKHS.

## 3 Regularized Risk Minimization in RKHS with Invariances

To simplify the presentation, we first describe our framework in the settings of semi-supervised learning [SSL, 4], and later show how to extend it to other learning scenarios in a straightforward way. In SSL, we wish to learn a target function $f$ both from labeled data and from local invariances extracted from labeled and unlabeled data. Let $(\mathbf{x}_1, y_1), \dots, (\mathbf{x}_l, y_l)$ be the labeled training data, and $\ell_1(f(\mathbf{x}), y)$ be the loss function on $f$ when the training input $\mathbf{x}$ is labeled as $y$. In this paper we restrict $\ell_1$ to be convex in its first argument, *e.g.* logistic loss, hinge loss, and squared loss.

We measure deviations from local invariances around each labeled or unlabeled input instance, and express them as bounded linear functionals $L_{l+1}(f), \dots, L_{l+m}(f)$ on the RKHS $\mathcal{H}$. The linear functionals are associated with another convex loss function $\ell_2(L_i(f))$ penalizing violations of the local invariances. As an example, the derivative of $f$ with respect to an input feature at some training instance $\mathbf{x}$ is a linear functional in $f$, and the loss function penalizes large values of the derivative at $\mathbf{x}$ using, *e.g.*, squared loss, absolute loss, and $\epsilon$-insensitive loss. Section 4 describes other local invariances we can consider and shows that these can be expressed as bounded linear functionals.

Finally, we penalize the complexity of $f$ via the squared RKHS norm $\|f\|^2$. Putting together the loss and regularizer, we set out to minimize the regularized risk functional over $f \in \mathcal{H}$:

$$\min_{f \in \mathcal{H}} \; \frac{1}{2}\|f\|^2 + \lambda \sum_{i=1}^{l} \ell_1(f(x_i), y_i) + \nu \sum_{i=l+1}^{l+m} \ell_2\left(L_i(f)\right), \quad (2)$$

where $\lambda, \nu > 0$. By the convexity of $\ell_1$ and $\ell_2$, (2) must be a convex optimization problem. However it is still in the function space and involves functionals. In order to derive an efficient optimization procedure, we now derive a representer theorem showing that the optimal solution lies in the span of a finite number of functions associated with the labeled data and the representers of the functionals $L_i$. Similar results are available in [22].

**Theorem 2.** *Let $\mathcal{H}$ be the RKHS defined by the kernel $k$. Let $L_i$ ($i = l+1, \dots, l+m$) be bounded linear functionals on $\mathcal{H}$ with representers $z_i$. Then the optimal solution to (2) must be in the form of*

$$g(\cdot) = \sum_{i=1}^{l} \alpha_i k(x_i, \cdot) + \sum_{i=l+1}^{l+m} \alpha_i z_i(\cdot) \quad (3)$$

*Furthermore, the parameters $\boldsymbol{\alpha} = (\alpha_1, \dots, \alpha_{l+m})'$ (finite dimensional) can be found by minimizing*

$$\lambda \sum_{i=1}^{l} \ell_1(\langle k(x_i, \cdot), f\rangle, y_i) + \nu \sum_{i=l+1}^{l+m} \ell_2\left(\langle z_i, f\rangle\right) + \frac{1}{2}\boldsymbol{\alpha}' K \boldsymbol{\alpha}, \text{ where } f = \sum_{i=1}^{l} \alpha_i k(x_i, \cdot) + \sum_{i=l+1}^{l+m} \alpha_i z_i. \quad (4)$$

*Here $K_{ij} = \langle \hat{k}_i, \hat{k}_j\rangle$, where $\hat{k}_i = k(x_i, \cdot)$ if $i \leq l$, and $\hat{k}_i = z_i(\cdot)$ otherwise. (Proof is in Appendix A.)*

Theorem 2 is similar to the results in [18, Proposition 3] and [19, Eq 8], where the optimal function lies in the span of a finite number of representers. However, our model is quite different in that it uses the representers of the linear functionals corresponding to the invariance, rather than virtual samples drawn from the invariant neighborhood. This could result in more compact models because the invariance (*e.g.* rotation) is enforced by a single representer, rather than multiple virtual examples (*e.g.* various degrees of rotation) drawn from the trajectory of invariant transforms. By the expansion of $f$ in (4), the labeling of a new instance $x$ depends not only on $k(x, x_i)$ that often measures the similarity between $x$ and training examples, but also takes into account the extent to which $k(x, \cdot)$, as a function, conforms to the prior invariances.

Computationally, $\langle k(x_i, \cdot), z_j \rangle$ is straightforward based on the definition of $L_j$. The efficient computation of $\langle z_i, z_j \rangle$ depends on the specific kernels and invariances, as we will show in Section 4. In the simplest case, consider the commonly used graph Laplacian regularizer [10, 11] which, given the similarity measure $w_{ij}$ between $x_i$ and $x_j$, can be written as $\sum_{ij} w_{ij}(f(x_i) - f(x_j))^2 = \sum_{ij} w_{ij}(L_{ij}(f))^2$, where $L_{ij}(f) = \langle f, k(x_i, \cdot) - k(x_j, \cdot) \rangle$ is linear and bounded. Then $\langle z_{ij}, z_{pq} \rangle = k(x_i, x_p) + k(x_j, x_q) - k(x_j, x_p) - k(x_i, x_q)$. Another generic approach is to use the assumption in Example 3 that $\langle z_s, f \rangle = L_s(f) = T_s(f)(x_s), \forall s \in \{i, j\}$. Then

$$\langle z_i, z_j \rangle = L_i(z_j) = T_i(z_j)(x_i) = \langle T_i(z_j), k(x_i, \cdot) \rangle = \langle z_j, T_i^*(k(x_i, \cdot)) \rangle = T_j(T_i^*(k(x_i, \cdot)))(x_j). \quad (5)$$

In practice, classifiers such as the support vector machine often use an additional constant term (bias) that is not penalized in the optimization. This is equivalent to searching $f$ over $\mathcal{F} + \mathcal{H}$, where $\mathcal{F}$ is a finite set of basis functions. A similar representer theorem can be established (see Appendix A).

## 4    Local Invariances as Bounded Linear Functionals

Interestingly, many useful local invariances can be modeled as bounded linear functionals. If it can be expressed in terms of function values $f(x)$, then it must be bounded as shown in Example 1.

In general, boundedness hinges on the functional and the RKHS $\mathcal{H}$. When $\mathcal{H}$ is finite dimensional, such as that induced by linear or polynomial kernels, all linear functionals on $\mathcal{H}$ must be bounded:

**Theorem 3** ([29, Thm 2.7-8]). *Linear functionals on finite dimensional normed space are bounded.*

However, in most nonparametric statistics problems that are of interest, the RKHS is infinite dimensional. So the boundedness requires a more refined analysis depending on the specific functional.

### 4.1    Differentiation Functional

In semi-supervised learning, a common prior is that the discriminant function $f$ does not change rapidly around sampled points. Therefore, we expect the norm of the gradient at these locations is small. Suppose $\mathcal{X} \subseteq \mathbb{R}^n$ is an open set, and $k$ is continuously differentiable on $\mathcal{X}^2$. Then we are interested in linear functionals $L_{\mathbf{x}_i, d}(f) := \frac{\partial f(\mathbf{x})}{\partial x^d}|_{\mathbf{x} = \mathbf{x}_i}$, where $x^d$ stands for the $d$-th component of the vector $\mathbf{x}$. Then $L_{\mathbf{x}_i, d}$ must be bounded:

**Theorem 4.** $L_{\mathbf{x}_i, d}$ *is bounded on* $\mathcal{H}$ *with respect to the RKHS norm.*

*Proof.* This result is immediate from the inequality given by [28, Corollary 4.36]:

$$\left| \frac{\partial}{\partial x^d} \Big|_{\mathbf{x}=\mathbf{x}_i} f(\mathbf{x}) \right| \leq \|f\| \left( \frac{\partial^2}{\partial x^d \partial y^d} \Big|_{\mathbf{x}=\mathbf{y}=\mathbf{x}_i} k(\mathbf{x}, \mathbf{y}) \right)^{\frac{1}{2}}. \qquad \blacksquare$$

Let us denote the representer of $L_{\mathbf{x}_i, d}$ as $z_{i,d}$. Indeed, [28, Corollary 4.36] established the same result for higher order partial derivatives, which can be easily used in our framework as well.

The inner product between representers can be computed by definition:

$$\langle z_{i,d}, z_{j,d'} \rangle = \frac{\partial}{\partial y^{d'}} \bigg|_{\mathbf{y}=\mathbf{x}_j} z_{i,d}(\mathbf{y}) = \frac{\partial}{\partial y^{d'}} \bigg|_{\mathbf{y}=\mathbf{x}_j} \langle z_{i,d}, k(\mathbf{y}, \cdot) \rangle = \frac{\partial^2}{\partial y^{d'} \partial x^d} \bigg|_{\mathbf{x}=\mathbf{x}_i, \mathbf{y}=\mathbf{x}_j} k(\mathbf{x}, \mathbf{y}). \quad (6)$$

If $k$ is considered as a function on $(\mathbf{x}, \mathbf{y}) \in \mathbb{R}^{2n}$, this implies that the inner product $\langle z_{i,d}, z_{j,d'} \rangle$ is the $(x^d, y^{d'})$-th element of the Hessian of $k$ evaluated at $(\mathbf{x}_i, \mathbf{x}_j)$, which could be interpreted as some sort of "covariance" between the two invariances with respect to the touchstone function $k$.

Applying (6) to the polynomial kernel $k(\mathbf{x}, \mathbf{y}) = (\langle \mathbf{x}, \mathbf{y} \rangle + 1)^r$, we derive

$$\langle z_{i,d}, z_{j,d'} \rangle = r(\langle \mathbf{x}_i, \mathbf{x}_j \rangle + 1)^{r-2}[(r-1)x_i^{d'} x_j^d + (\langle \mathbf{x}_i, \mathbf{x}_j \rangle + 1)\delta_{d=d'}], \qquad (7)$$

where $\delta_{d=d'} = 1$ if $d = d'$, and 0 otherwise. For Gaussian kernels $k(\mathbf{x}, \mathbf{y}) = \kappa_\sigma(\mathbf{x}, \mathbf{y})$, we can take another path that is different from (6). Note that $L_{\mathbf{x}_i, d}(f) = T(f)(\mathbf{x}_i)$ where $T : f \mapsto \frac{\partial f}{\partial x^d}$, and it is straightforward to verify that $T$ is bounded with $T^* = -T$ for Gaussian RKHS. So applying (5),

$$\langle z_{i,d}, z_{j,d'} \rangle = \left. \frac{\partial}{\partial y^{d'}} \right|_{\mathbf{y} = \mathbf{x}_j} \left( -\frac{\partial}{\partial y^d} k(\mathbf{x}_i, \mathbf{y}) \right) = \frac{k(\mathbf{x}_i, \mathbf{x}_j)}{\sigma^4} [\sigma^2 \delta_{d=d'} - (x_i^d - x_j^d)(x_i^{d'} - x_j^{d'})]. \quad (8)$$

By Theorem 1, it immediately follows that the norm of $L_{\mathbf{x}_i, d}$ is $\sqrt{\langle z_{i,d}, z_{i,d} \rangle} = 1/\sigma$.

## 4.2   Transformation Invariance

Invariance to known local transformations of input has been used successfully in supervised learning [3]. Here we show that transformation invariance can be handled in our framework via representers in RKHS. In particular, gradients with respect to the transformations are bounded linear functionals.

Following [3], we first require a differentiable function $g$ that maps points from a space $\mathcal{S}$ to $\mathbb{R}$, where $\mathcal{S}$ lies in a Euclidean space.[3] For example, an image can be considered as a function $g$ that maps points in the plane $\mathcal{S} = \mathbb{R}^2$ to the intensity of the image at that point. Next, we consider a family of bijective transformations $t_{\boldsymbol{\alpha}} : \mathcal{S} \mapsto \mathcal{S}$, which is differentiable in both the input and the parameter $\boldsymbol{\alpha}$. For instance, translation, rotation, and scaling can be represented as mappings $t_{\boldsymbol{\alpha}}^{(1)}$, $t_\alpha^{(2)}$, and $t_\alpha^{(3)}$ respectively:

$$\begin{pmatrix} x \\ y \end{pmatrix} \overset{t_{\boldsymbol{\alpha}}^{(1)}}{\longmapsto} \begin{pmatrix} x + \alpha_x \\ y + \alpha_y \end{pmatrix}, \quad \begin{pmatrix} x \\ y \end{pmatrix} \overset{t_\alpha^{(2)}}{\longmapsto} \begin{pmatrix} x \cos\alpha - y \sin\alpha \\ x \sin\alpha - y \cos\alpha \end{pmatrix}, \quad \begin{pmatrix} x \\ y \end{pmatrix} \overset{t_\alpha^{(3)}}{\longmapsto} \begin{pmatrix} x + \alpha x \\ y + \alpha y \end{pmatrix}.$$

Based on $t_{\boldsymbol{\alpha}}$, we define a family of operators $T_{\boldsymbol{\alpha}} : \mathbb{R}^{\mathcal{S}} \to \mathbb{R}^{\mathcal{S}}$ as $T_{\boldsymbol{\alpha}}(g) = g \circ t_{\boldsymbol{\alpha}}^{-1}$. The function $T_{\boldsymbol{\alpha}}(g)(x, y)$ gives us the intensity at location $(x, y)$ of the image translated by an offset $(\alpha_x, \alpha_y)$, rotated by an angle $\alpha$, or scaled by an amount $\alpha$. Finally we sample from $\mathcal{S}$ a fixed number of locations $S := \{\mathbf{s}_1, \ldots, \mathbf{s}_q\}$, and present to the learning algorithm a vector $I(g, \boldsymbol{\alpha}; S) := (T_{\boldsymbol{\alpha}}(g)(\mathbf{s}_1)', \ldots, T_{\boldsymbol{\alpha}}(g)(\mathbf{s}_q)')'$. Digital images are discretization of real images where we sample at fixed pixel locations of the function $g$ to obtain a fixed sized vector. Clearly, for a fixed $g$, the sampled observation $I(g, \boldsymbol{\alpha}; S)$ is a vector valued function of $\boldsymbol{\alpha}$.

The following result allows our framework to use derivatives with respect to the parameters in $\boldsymbol{\alpha}$.

**Theorem 5.** *Let $\mathcal{F}$ be a normed vector space of functions that map from the range of $I(g, \boldsymbol{\alpha}; S)$ to $\mathbb{R}$. Suppose the linear functional that maps $f \in \mathcal{F}$ to $\left. \frac{\partial f(\mathbf{u})}{\partial u^j} \right|_{\mathbf{u} = \mathbf{u}_0}$ is bounded for any $\mathbf{u}_0$ and coordinate $j$, and its norm is denoted as $C_j$. Then the functional $L_{g,d,S}$: $f \mapsto \left. \frac{\partial}{\partial \alpha^d} \right|_{\boldsymbol{\alpha} = \mathbf{0}} f(I(g, \boldsymbol{\alpha}; S))$, i.e. derivatives with respect to each of the components in $\boldsymbol{\alpha}$, must be bounded linear functionals on $\mathcal{F}$.*

*Proof.* Let $I^j(g, \boldsymbol{\alpha}; S)$ be the $j$-th component of $I(g, \boldsymbol{\alpha}; S)$. Using the chain rule, for any $f \in \mathcal{F}$

$$|L_{g,d,S}(f)| = \left| \left. \frac{\partial f(I(g, \boldsymbol{\alpha}; S))}{\partial \alpha^d} \right|_{\boldsymbol{\alpha} = \mathbf{0}} \right| = \left| \sum_{j=1}^q \left. \frac{\partial f(\mathbf{u})}{\partial u^j} \right|_{\mathbf{u} = I(g, \mathbf{0}; S)} \cdot \left. \frac{\partial I^j(g, \boldsymbol{\alpha}; S)}{\partial \alpha^d} \right|_{\boldsymbol{\alpha} = \mathbf{0}} \right|$$

(by definition of $\|f\|$) $\displaystyle \leq \sum_{j=1}^q (C_j \|f\|) \cdot \left| \left. \frac{\partial I^j(g, \boldsymbol{\alpha}; S)}{\partial \alpha^d} \right|_{\boldsymbol{\alpha} = \mathbf{0}} \right| = \|f\| \cdot \sum_{j=1}^q C_j \left| \left. \frac{\partial I^j(g, \boldsymbol{\alpha}; S)}{\partial \alpha^d} \right|_{\boldsymbol{\alpha} = \mathbf{0}} \right|.$

The proof is completed by noting that the last summation is a finite constant independent of $f$. ∎

**Corollary 1.** *The derivatives $\left. \frac{\partial}{\partial \alpha^d} \right|_{\boldsymbol{\alpha} = \mathbf{0}} f(I(g, \boldsymbol{\alpha}; S))$ with respect to each of the component in $\boldsymbol{\alpha}$ are bounded linear functionals on the RKHS defined by the polynomial and Gaussian kernels.*

To compute the inner product between representers, let $z_{g,d,S}$ be the representer of $L_{g,d,S}$ and denote $\mathbf{v}_{g,d,S} = \frac{\partial I(g,\boldsymbol{\alpha};S)}{\partial \alpha^d}|_{\boldsymbol{\alpha} = \mathbf{0}}$. Then $\langle k(\mathbf{x}, \cdot), z_{g,d,S} \rangle = \left\langle \left. \frac{\partial}{\partial \mathbf{y}} \right|_{\mathbf{y} = I(g, \mathbf{0}, S)} k(\mathbf{x}, \mathbf{y}), \mathbf{v}_{g,d,S} \right\rangle$ and

$$\langle z_{g,d,S}, z_{g',d',S'} \rangle = \left\langle \mathbf{v}_{g,d,S}, \left. \frac{\partial}{\partial \mathbf{x}} \right|_{\mathbf{x} = I(g, \mathbf{0}, S)} \left( \left\langle \mathbf{v}_{g',d',S'}, \left. \frac{\partial}{\partial \mathbf{y}} \right|_{\mathbf{y} = I(g', \mathbf{0}, S')} k(\mathbf{x}, \mathbf{y}) \right\rangle \right) \right\rangle. \quad (9)$$

### 4.3 Local Averaging

Using gradients to enforce the local invariance that the target function does not change much around data instances increases the number of basis functions by a factor of $n$, where $n$ is the number of gradient directions that we use. The optimization problem can become computationally expensive if $n$ is large. When we do not have useful information about the invariant directions, it may be useful to have methods that do not increase the number of basis functions by much. Consider functionals

$$L_{\mathbf{x}_i}(f) = \int_{\mathcal{X}} f(\boldsymbol{\tau}) p(\mathbf{x}_i - \boldsymbol{\tau}) \mathrm{d}\boldsymbol{\tau} - f(\mathbf{x}_i), \tag{10}$$

where $p(\cdot)$ is a probability density function centered at zero. Minimizing a loss with such linear functionals will favor functions whose local averages given by the integral are close to the function values at data instances. If $p(\cdot)$ is selected to be a low pass filter, the function should be smoother and less likely to change in regions with more data points but is less constrained to be smooth in regions where the data points are sparse. Hence, such loss functions may be appropriate when we believe that data instances from the same class are clustered together.

To use the framework we have developed, we need to select the probability density $p(\cdot)$ and the kernel $k$ such that $L_{\mathbf{x}_i}(f)$ is a bounded linear functional.

**Theorem 6.** *Assume there is a constant $C > 0$ such that $k(\mathbf{x}, \mathbf{x}) \leq C^2$ for all $\mathbf{x} \in \mathcal{X}$. Then the linear functional $L_{\mathbf{x}_i}$ in* (10) *is bounded in the RKHS defined by $k$ for any probability density $p(\cdot)$.*

See proof in Appendix B. As a result, radial kernels such as Gaussian and exponential kernels make $L_{\mathbf{x}_i}$ bounded. $k(\mathbf{x}, \mathbf{x})$ is not bounded for polynomial kernel, but it has been covered by Theorem 3.

To allow for efficient implementation, the inner product between representers of invariances must be computed efficiently. Unlike differentiation, integration often does not result in a closed form expression. Fortunately, analytic evaluation is feasible for the Gaussian kernel $\kappa_\sigma(\mathbf{x}_i, \mathbf{x}_j)$ together with the Gaussian density function $p(\mathbf{x}) = (\theta\sqrt{2\pi})^{-n} \kappa_\theta(\mathbf{x}, \mathbf{0})$, because the convolution of two Gaussian densities is still a Gaussian density (see derivation in Appendix B):

$$\langle z_{\mathbf{x}_i}, z_{\mathbf{x}_j} \rangle = \kappa_\sigma(\mathbf{x}_i, \mathbf{x}_j) + (1 + 2\theta/\sigma)^{-n} \kappa_{\sigma+2\theta}(\mathbf{x}_i, \mathbf{x}_j) - 2(1 + \theta/\sigma)^{-n} \kappa_{\sigma+\theta}(\mathbf{x}_i, \mathbf{x}_j).$$

## 5 Optimization

The objective (2) can be optimized by many algorithms, such as stochastic gradient [30], bundle method [19, 31], and (randomized) coordinate descent in its dual (4) [32]. Since all computational strategies rely on kernel evaluation, we prefer the dual approach. In particular, (4) allows an unconstrained optimization and the nonsmooth regions in $\ell_1$ or $\ell_2$ can be easily approximated by smooth surrogates [33]. So without loss of generality, we assume $\ell_1$ and $\ell_2$ are smooth. Our approach can work in both batch and coordinate-wise, depending on the scale of different problems.

In the batch setting, the major challenge is the cost in computing the gradient when the number of invariance is large. For example, consider all derivatives at all labeled examples $\mathbf{x}_1, \ldots, \mathbf{x}_l$ and let $N = (\langle z_{i,d'}, z_{j,d'} \rangle)_{(i,d),(j,d')} \in \mathbb{R}^{nl \times nl}$ as in (8). Then given $\boldsymbol{\alpha} = (\boldsymbol{\alpha}_1', \ldots, \boldsymbol{\alpha}_l')' \in \mathbb{R}^{nl}$, the bottleneck of computation is $\mathbf{g} := N\boldsymbol{\alpha}$, which costs $O(l^2 n^2)$ time and $O(l^2 n^2)$ space to store $N$ in a vanilla implementation. However, since the kernel matrices often employ rich structures, a careful treatment can reduce the cost by an order of magnitude, *e.g.* into $O(l^2 n)$ time and $O(nl + l^2)$ space in this case. Specifically, denote $K = (k(\mathbf{x}_i, \mathbf{x}_j))_{ij} \in \mathbb{R}^{l \times l}$, and three $n \times l$ matrices $X = (\mathbf{x}_1, \ldots, \mathbf{x}_l)$, $A = (\boldsymbol{\alpha}_1, \ldots, \boldsymbol{\alpha}_l)$, and $G = (g_{d,i})$. Then one can show

$$G = \sigma^{-4} \left[ \sigma^2 AK + XQ - X \circ (\mathbf{1}_n \otimes \mathbf{1}_l' Q) \right], \quad \text{where } Q_{ij} = K_{ij}(\Lambda_{ji} - \Lambda_{ii}), \ \Lambda = X'A. \tag{11}$$

Here $\circ$ stands for Hadamard product, and $\otimes$ is Kronecker product. $\mathbf{1}_n \in \mathbb{R}^n$ is a vector of straight one. The computational cost is dominated by $X'A$, $AK$, and $XQ$, which are all $O(l^2 n)$.

When the number of invariance is huge, a batch solver can be slow and coordinate-wise updates can be more efficient. In each iteration, it picks a coordinate in $\boldsymbol{\alpha}$ and optimizes the objective over all the coordinates picked so far, leaving the rest elements to zero. [32] selected the coordinate randomly, while another strategy is to choose the steepest descent coordinate. Clearly, it is useful only when this selection can be performed efficiently, which depends heavily on the structure of the problem.

## 6 Experimental Result

We compared our approach, which is henceforth referred to as InvSVM, with state-of-the-art methods

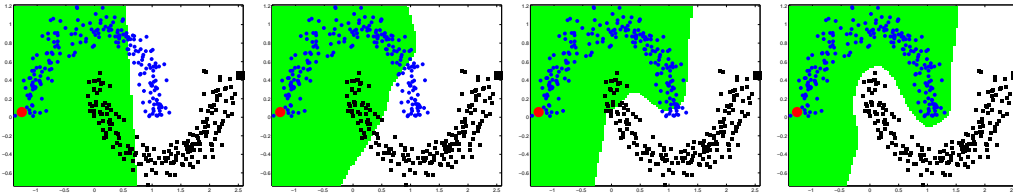

Figure 1: Decision boundary for the two moon dataset with $\nu = 0, 0.01, 0.1$, and $1$ (left to right).

in (semi-)supervised learning using invariance to differentiation and transformation.

## 6.1 Invariance to Differentiation: Transduction on Two Moon Dataset

As a proof of concept, we experimented with the "two moon" dataset shown in Figure 1, with only $l = 2$ labeled data instances (red circle and black square). We used the Gaussian kernel with $\sigma = 0.25$ and the gradient invariances on *both labeled and unlabeled data*. The loss $\ell_1$ and $\ell_2$ are logistic and squared loss respectively, with $\lambda = 1$, and $\nu \in \{0, 0.01, 0.1, 1\}$. (4) was minimized by a L-BFGS solver [34]. In Figure 1, from left to right our method lays more and more emphasis on placing the separation boundary in low density regions, which allows unlabeled data to improve the classification accuracy. We also tried hinge loss for $\ell_1$ and $\epsilon$-insensitive loss for $\ell_2$ with similar classification results.

## 6.2 Invariance to Differentiation: Semi-supervised Learning on Real-world Data.

**Datasets.** We used 9 datasets for binary classification from [4] and the UCI repository [35]. The number of features ($n$) and instances ($t$) are given in Table 1. All feature vectors were normalized to zero mean and unit length.

**Algorithms.** We trained InvSVM with hinge loss for $\ell_1$ and squared loss for $\ell_2$. The differentiation invariance was used over the whole dataset, *i.e.* $m = nt$ in (4), and the gradient computation was accelerated by (11). We compared InvSVM with the standard SVM, and a state-of-the-art semi-supervised learning algorithm LapSVM [36], which uses manifold regularization based on graph Laplacian [11, 37]. All three algorithms used Gaussian kernel with the bandwidth $\sigma$ set to be the median of pairwise distance among all instances.

**Settings.** We trained SVM, LapSVM, and InvSVM on a subset of $l \in \{30, 60, 90\}$ labeled examples, and compared their test error on the other $t - l$ examples in each dataset. We used 5 fold stratified cross validation (CV) to select the values of $\lambda$ and $\nu$ for InvSVM. CV was also applied to LapSVM for choosing the number of nearest neighbor for graph construction, weight on the Laplacian regularizer, and the standard RKHS norm regularizer. For each fold of CV, InvSVM and LapSVM used the other 4 folds ($\frac{4}{5}l$ points) as labeled data, and the remaining $t - \frac{4}{5}l$ points as unlabeled data. The error on the $\frac{1}{5}l$ points was then used for CV. Finally the random selection of $l$ labeled examples was repeated for 10 times, and we reported the mean test error and standard deviation.

**Results.** It is clear from Table 1 that in most cases InvSVM achieves lower or similar test error compared to SVM and LapSVM. Both LapSVM and InvSVM are implementations of the low density prior. To this end, LapSVM enforces smoothness of the discrimination function *over neighboring instances*, while InvSVM directly penalizes the gradient at instances and does not require a notion of neighborhood. The lower error of InvSVM suggests the superiority of the use of gradient, which is enabled by our represener based approach. Besides, SVM often performs quite well when the number of labeled data is large, with similar error as LapSVM. But still, InvSVM can attain even lower error.

## 6.3 Transformation Invariance

Next we study the use of transformation invariance for supervised learning. We used the handwritten digits from the MNIST dataset [38] and compared InvSVM with the virtual sample SVM (VirSVM) which constructs additional instances by applying the following transformations to the training data: 2-pixel shifts in 4 directions, rotations by $\pm 10$ degrees, scaling by $\pm 0.1$ unit, and shearing in vertical or horizontal axis by $\pm 0.1$ unit. For InvSVM, the derivative $\frac{\partial}{\partial \boldsymbol{\alpha}}|_{\boldsymbol{\alpha}=0} I(g, \boldsymbol{\alpha}; S)$ was approximated with the difference that results from the above transformation. We considered binary classification problems by choosing four pairs of digits (4-vs-9, 2-vs-3, and 6-vs-5 are hard, while 7-vs-1 is

Table 1: Test error of SVM, LapSVM, and InvSVM for semi-supervised learning. The best result (including tie) that is statistically significant in each setting is highlighted in bold. No number is highlighted if there is no significant difference between the three methods.

| Method | **heart** ($n = 13, t = 270$) | | | **BCI** ($n = 117, t = 400$) | | | **bupa** ($n = 6, t = 245$) | | |
|---|---|---|---|---|---|---|---|---|---|
| | $l = 30$ | $l = 60$ | $l = 90$ | $l = 30$ | $l = 60$ | $l = 90$ | $l = 30$ | $l = 60$ | $l = 90$ |
| SVM | 23.5±2.08 | 21.4±0.23 | **20.0±1.11** | 44.6±1.89 | **39.0±3.41** | 34.6±3.25 | **36.2±5.53** | 37.9±5.80 | 36.9±1.80 |
| LapSVM | 23.2±1.68 | 22.7±1.92 | **20.7±2.10** | 44.8±2.72 | 45.4±2.25 | 36.1±3.92 | **36.6±5.98** | 40.7±4.05 | 37.1±1.58 |
| InvSVM | **22.3±1.27** | **20.2±1.01** | 19.6±2.79 | 45.3±3.59 | **38.4±4.41** | 32.7±2.27 | 38.2±4.46 | **35.4±1.85** | **35.2±0.45** |
| | **g241c** ($n = 241, t = 1500$) | | | **g241n** ($n = 241, t = 1500$) | | | **Australian** ($n = 14, t = 690$) | | |
| | $l = 30$ | $l = 60$ | $l = 90$ | $l = 30$ | $l = 60$ | $l = 90$ | $l = 30$ | $l = 60$ | $l = 90$ |
| SVM | **32.9±1.59** | 27.1±0.92 | 24.8±1.99 | **34.6±2.52** | **28.3±2.65** | 25.6±1.48 | 20.6±4.18 | 21.7±11.3 | 15.3±0.86 |
| LapSVM | 37.1±1.25 | 28.4±2.44 | 29.7±1.57 | 37.7±5.76 | 29.9±2.41 | 25.9±1.49 | 21.9±9.27 | **15.6±2.49** | **14.7±0.16** |
| InvSVM | **33.1±0.49** | **26.4±1.14** | **23.4±1.36** | 35.4±1.57 | **28.4±3.03** | 22.3±4.69 | **17.7±1.74** | **16.4±1.11** | 15.6±0.77 |
| | **ionosphere** ($n = 34, t = 351$) | | | **sonar** ($n = 60, t = 208$) | | | **USPS** ($n = 241, t = 1500$) | | |
| | $l = 30$ | $l = 60$ | $l = 90$ | $l = 30$ | $l = 60$ | $l = 90$ | $l = 30$ | $l = 60$ | $l = 90$ |
| SVM | 12.5±3.12 | 8.71±1.62 | **7.17±1.67** | 30.9±2.53 | **22.7±0.39** | 20.6±4.81 | 14.9±0.26 | 12.6±2.04 | 11.3±2.06 |
| LapSVM | 14.9±1.73 | 9.05±1.05 | 7.66±1.53 | **29.4±2.33** | 22.9±0.17 | 24.9±1.29 | 15.3±1.10 | 12.3±1.74 | 11.1±1.81 |
| InvSVM | **7.58±1.29** | **7.90±0.23** | 7.02±0.88 | 31.6±4.68 | 24.1±2.06 | **21.8±3.91** | 15.4±4.02 | 12.2±2.93 | 11.3±1.63 |

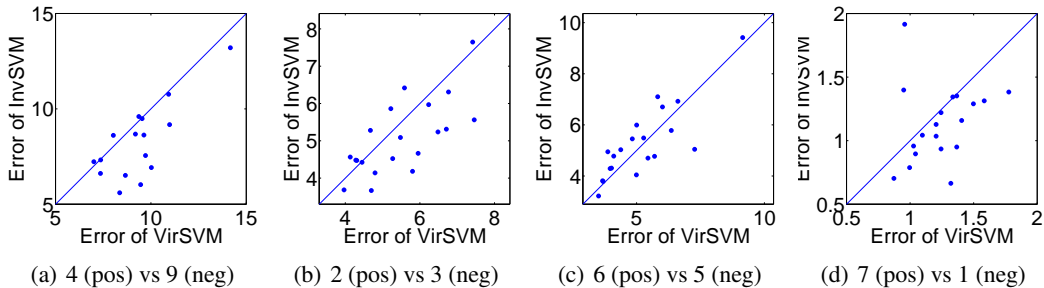

(a) 4 (pos) vs 9 (neg)     (b) 2 (pos) vs 3 (neg)     (c) 6 (pos) vs 5 (neg)     (d) 7 (pos) vs 1 (neg)

Figure 2: Test error (in percentage) of InvSVM versus SVM with virtual sample.

easier). As the real distribution of digit is imbalanced and invariance is more useful when the number of labeled data is low, we randomly chose $n_+ = 50$ labeled images for one class and $n_- = 10$ images for the other. Accordingly the supervised loss was normalized within each class: $n_+^{-1} \sum_{i:y_i=1} \ell_1(f(\mathbf{x}_i), 1) + n_-^{-1} \sum_{i:y_i=-1} \ell_1(f(\mathbf{x}_i), -1)$. Logistic loss and $\epsilon$-insensitive loss were used for $\ell_1$ and $\ell_2$ respectively. All parameters were set by 5 fold CV and the test error was measured on the rest images in the dataset. The whole process was repeated for 20 times.

In Figure 2, InvSVM generally yields lower error than VirSVM, which suggests that compared with drawing virtural samples from the invariance, it seems more effective to directly enforce a flat gradient like in [3].

## 7 Conclusion and Discussion

We have shown how to model local invariances by using represeners in RKHS. This subsumes a wide range of invariances that are useful in practice, and the formulation can be optimized efficiently.

For future work, it will be interesting to extend the framework to a broader range of learning tasks. A potential application is to the problem of imbalanced data learning, where one wishes to keep the decision boundary further away from instances of the minority class and closer to the instances of majority class. It will also be interesting to generalize the framework to invariances that are not directly linear. For example, the total variation $\ell(f) := \int_a^b |f'(x)| \, \mathrm{d}x$ ($a, b \in \mathbb{R}$) is not linear, but we can treat the integral of absolute value as a loss function and take the derivative as the linear functional: $L_x(f) = f'(x)$ and $\ell(f) = \int_a^b |L_x(f)| \, \mathrm{d}x$. As a result, the optimization may have to resort to sparse approximation methods. Finally the norm of the linear functional does affect the optimization efficiency, and a detailed analysis will be useful for choosing the linear functionals.

## Footnotes

[1]A similar result was provided in [22], but not in the context of learning with invariance.

[2]We write a variable in boldface if it is a vector in a Euclidean space.

[3]In practice, $\mathcal{S}$ can be a discrete domain such as the pixel coordinate of an image. Then $g$ can be extended to an interpolated continuous space via convolution with a Gaussian. See [3, § 2.3] for more details.

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
