[Supplementary Material · appendix.pdf]

# Supplementary Material for *Learning with Invariance via Linear Functionals on Reproducing Kernel Hilbert Space*

## A   Proof of Representer Theorem 2

*Proof.* By Riesz's Theorem, the linear functional $L_i(f)$ can be represented as an inner product $L_i(f) = \langle f, z_i \rangle$, where $z_i$ is a member of the Hilbert space and $z_i(x) = \langle z_i, k(x, \cdot) \rangle$.

We now claim that the solution of the optimization problem (2) can be represented as

$$\sum_{i=1}^{l} \alpha_i k(x_i, \cdot) + \sum_{i=l+1}^{n} \alpha_i z_i(\cdot).$$

To see this, any function $f$ in the RKHS can be represented as

$$f(\cdot) = \sum_{i=1}^{l} \alpha_i k(x_i, \cdot) + \sum_{i=l+1}^{n} \alpha_i z_i(\cdot) + f_\perp(\cdot)$$

where $f_\perp(x)$ is in the orthogonal complement of the span of $k(x_i, \cdot)$ for $1 \leq i \leq l$ and of $z_i(\cdot)$ for $l + 1 \leq i \leq n$. Each of the terms that contains the loss function $l_2$ depends only on $f(x_k)$ which can be written as

$$
\begin{aligned}
f(x_k) &= \langle f(\cdot), k(x_k, \cdot) \rangle \\
&= \sum_{i=1}^{l} \alpha_i \langle k(x_i, \cdot), k(x_k, \cdot) \rangle + \sum_{i=l+1}^{n} \alpha_i \langle z_i(\cdot), k(x_k, \cdot) \rangle + \langle f_\perp(x), k(x_k, \cdot) \rangle \\
&= \sum_{i=1}^{l} \alpha_i k(x_i, x_k) + \sum_{i=l+1}^{n} \alpha_i z_i(x_k).
\end{aligned}
$$

Hence, each of those terms depends only on the projection of the function onto the span.

Similarly, each of the terms that contain the loss function $l_1$ depends only on the projection of the function onto the span.

$$
\begin{aligned}
L_k(f) = \langle f, z_k \rangle &= \sum_{i=1}^{l} \alpha_i \langle k(x_i, \cdot), z_k \rangle + \sum_{i=l+1}^{n} \alpha_i \langle z_i, z_k \rangle + \langle f_\perp(x), z_k \rangle \\
&= \left\langle \sum_{i=1}^{l} \alpha_i k(x_i, \cdot) + \sum_{i=l+1}^{n} \alpha_i z_i, z_k \right\rangle.
\end{aligned}
$$

Since all quantities in the loss functions $l_1$ and $l_2$ depend only on the component that lies in the span, any function that has components in the orthogonal complement has higher cost than its projection onto the span. Hence the optimal solution must lie in the span of the desired functions.

Finally, note that the squared norm of the function

$$f(\cdot) = \sum_{i=1}^{l} \alpha_i k(x_i, \cdot) + \sum_{i=l+1}^{n} \alpha_i z_i(\cdot)$$

can be written as $\boldsymbol{\alpha}' K \boldsymbol{\alpha}$ where $K_{ij} = \langle k_i', k_j' \rangle$ where $k_i' = k(x_i, \cdot)$ if $i \leq l$ and $k_i' = z_i(\cdot)$ otherwise. ∎

The following more general version of the representer theorem covers this case as well. The proof is similar to that of Theorem 2.

**Theorem 7.** *Let $L_i$, $i = l + 1, \ldots, n$, be bounded linear functionals in the reproducing kernel Hilbert space $\mathcal{H}$ defined by the kernel $k$. Let $\mathcal{F}$ be the span of a fixed set of basis functions $\phi_j$, $j = 1, \ldots, m$. The solution of the optimization problem*

$$g = \underset{f=f_1+f_2, f_1 \in \mathcal{F}, f_2 \in \mathcal{H}}{\operatorname{argmin}} \lambda \sum_{i=1}^{l} l_2(y_i, f(x_i)) + \nu \sum_{i=l+1}^{n} l_1\left(L_i(f)\right) + \frac{1}{2}\|f_2\|^2$$

*for $\lambda, \nu > 0$ can be expressed as*

$$g(\cdot) = \sum_{j=1}^{m} w_j \phi_j(\cdot) + \sum_{i=1}^{l} \alpha_i k(x_i, \cdot) + \sum_{i=l+1}^{n} \alpha_i z_i(\cdot)$$

*where $z_i$ is the representer of $L_i$. Furthermore, the parameters $\mathbf{w} = (w_1, \ldots, w_m)'$ and $\boldsymbol{\alpha} = (\alpha_1, \ldots, \alpha_n)'$ can be obtained by minimizing*

$$\lambda \sum_{i=1}^{l} l_2(y_i, f(x_i)) + \nu \sum_{i=l+1}^{n} l_1\left(L_i(f)\right) + \frac{1}{2}\boldsymbol{\alpha}' K \boldsymbol{\alpha}$$

*where $f = \sum_{j=1}^{m} w_j \phi_j(\cdot) + \sum_{i=1}^{l} \alpha_i k(x_i, \cdot) + \sum_{i=l+1}^{n} \alpha_i z_i(\cdot)$ and $K_{i,j} = \langle k_i', k_j' \rangle$ where $k_i' = k(x_i, \cdot)$ if $i \le l$ and $k_i' = z_i(\cdot)$ otherwise.*

# B    Inner Product of Representers for Local Average

*Proof for Theorem 6.* By the reproducing kernel property and Cauchy-Schwarz inequality,

$$|f(\mathbf{x})| = |\langle f, k(\mathbf{x}, \cdot)\rangle| \le \|f\| \cdot \|k(\mathbf{x}, \cdot)\| = \|f\|\, k(\mathbf{x}, \mathbf{x})^{1/2} \le C\, \|f\|. \tag{12}$$

Therefore,    $|L_{\mathbf{x}_i}(f)| = \left| \int_{\mathcal{X}} f(\boldsymbol{\tau}) p(\mathbf{x}_i - \boldsymbol{\tau}) \mathrm{d}\boldsymbol{\tau} - f(\mathbf{x}_i) \right| \le \int_{\mathcal{X}} |f(\boldsymbol{\tau})| p(\mathbf{x}_i - \boldsymbol{\tau}) \mathrm{d}\boldsymbol{\tau} + C\, \|f\|$

$$\le C\, \|f\| \int_{\mathcal{X}} p(\mathbf{x}_i - \boldsymbol{\tau}) \mathrm{d}\boldsymbol{\tau} + C\, \|f\| = 2C\|f\|. \qquad \blacksquare$$

## B.1    Inner Products of Representers for Local Average Functional

Let $p(\mathbf{x}) = (\theta\sqrt{2\pi})^{-n} \kappa_\theta(\mathbf{x}, \mathbf{0})$. Then

$$\langle z_{\mathbf{x}_i}, k(\mathbf{x}, \cdot)\rangle = L_{\mathbf{x}_i}(k(\mathbf{x}, \cdot)) = \int_{\mathcal{X}} k(\mathbf{x}, \boldsymbol{\tau}) p(\mathbf{x}_i - \boldsymbol{\tau}) \mathrm{d}\boldsymbol{\tau} - k(\mathbf{x}_i, \mathbf{x})$$

$$= (2\pi)^{n/2}\sigma^n \int_{\mathcal{X}} \frac{1}{(2\pi)^{n/2}\sigma^n} \exp\left(-\frac{\|\mathbf{x} - \boldsymbol{\tau}\|^2}{2\sigma^2}\right) \frac{1}{(2\pi)^{n/2}\theta^n} \exp\left(-\frac{\|\mathbf{x}_i - \boldsymbol{\tau}\|^2}{2\theta^2}\right) \mathrm{d}\boldsymbol{\tau}$$

$$\qquad - k(\mathbf{x}_i, \mathbf{x})$$

$$= \frac{\sigma^n}{(\sigma + \theta)^n} \exp\left(-\frac{1}{2(\sigma + \theta)^2}\|\mathbf{x}_i - \mathbf{x}\|^2\right) - \exp\left(-\frac{1}{2\sigma^2}\|\mathbf{x}_i - \mathbf{x}\|^2\right)$$

$$= (1 + \theta/\sigma)^{-n} \kappa_{\sigma+\theta}(\mathbf{x}_i, \mathbf{x}) - \kappa_\sigma(\mathbf{x}, \mathbf{x}_i).$$

Similarly,

$$\langle z_{\mathbf{x}_i}, z_{\mathbf{x}_j} \rangle = L_{\mathbf{x}_i}(z_{\mathbf{x}_j}) = (2\pi)^{n/2}\sigma^n \int_{\mathcal{X}} \left[ \frac{1}{(2\pi)^{n/2}(\sigma + \theta)^n} \exp\left(-\frac{1}{2(\sigma + \theta)^2}\|\mathbf{x}_j - \mathbf{x}\|^2\right) \right.$$

$$\left. - \frac{1}{(2\pi)^{n/2}\sigma^n} \exp\left(-\frac{1}{2\sigma^2}\|\mathbf{x}_j - \mathbf{x}\|^2\right) \right]$$

$$\frac{1}{(2\pi)^{n/2}\theta^n} \exp\left(-\frac{1}{2\theta^2}\|\mathbf{x}_i - \boldsymbol{\tau}\|^2\right) \mathrm{d}\boldsymbol{\tau} - z_{\mathbf{x}_j}(\mathbf{x}_i)$$

$$= \frac{\sigma^n}{(\sigma + 2\theta)^n} \exp\left(-\frac{1}{2(\sigma + 2\theta)^2}\|\mathbf{x}_i - \mathbf{x}_j\|^2\right)$$

$$- \frac{\sigma^n}{(\sigma + \theta)^n} \exp\left(-\frac{1}{2(\sigma + \theta)^2}\|\mathbf{x}_i - \mathbf{x}_j\|^2\right) - z_{\mathbf{x}_j}(\mathbf{x}_i)$$

$$= \kappa_\sigma(\mathbf{x}_i, \mathbf{x}_j) + (1 + 2\theta/\sigma)^{-n}\kappa_{\sigma+2\theta}(\mathbf{x}_i, \mathbf{x}_j) - 2(1 + \theta/\sigma)^{-n}\kappa_{\sigma+\theta}(\mathbf{x}_i, \mathbf{x}_j).$$