[Reviews · NeurIPS 2013]

Submitted by Assigned_Reviewer_4

This paper deals with taking into account local invariances in RKHS (or GP) based classification. The approach is based on integrating the invariances as extra observations in the model instead of designing kernels encoding the invariance properties. The method is illustrated on both simulated and real world data.

The proposed method is well motivated and the paper is enjoyable to read. Although incorporating informations such as derivatives in the model is not new, the presentation given by the authors offers a coherent framework that also covers less trivial cases such as translation, rotation or scaling invariances. Furthermore, the various examples of the benchmark convinced me of the usefulness of the method.

From my point of view the overall method proposed by the authors is original and could benefit the NIPS community.

Minor remarks:
- One term is missing in the right hand side of Eq. 6
- One word is missing on page 8, line 424
Summary: The paper shows that encoding local invariances into the models as observations in a regularisation framework can reduce significantly the error rate in classification. The paper is theoretically sound and could benefit the NIPS community.

Submitted by Assigned_Reviewer_5

The paper presents an approach to encode local invariances around data instances in the framework of RKHSs. The authors give a representer theorem and discuss how to encode a number of invariances that can be used in applications. Finally they present experiments in which the proposed approaches compare favorably to existing alternatives.

The paper is overall well written and interesting. The representer theorem is closely related to alternative versions found in the literature; nonetheless the application to invariances, albeit not entirely new, is well developed (given the conference format) and tested by the authors.

minor comments

sec 4.2: why is alpha the same for all the transformations in the first formula? each transformation can have its own parameter, right? This is not clear in the formula.

Summary: The paper is well written and interesting. In comparison to the top NIPS paper the technical contribution is relatively limited; however experiments are convincing and, overall, well documented.

Submitted by Assigned_Reviewer_6

The authors address the interesting and important topic of learning with invariance.
As far as I know the proposed method is new, but related ideas have been published earlier.
The proposal relies on functional analysis, mainly the Riesz representation theorem and
reproducing kernel Hilbert spaces.
One main advantage of the proposal is a numerical one: one has to solve "only" a certain convex program. In contrast to classical SVMs, one has at least 2 hyperparameters plus kernel parameters, if present.

The authors derive a nice empirical representer theorem (Thm.2) and apply their method to some benchmark data sets.
Summary: Interesting idea. Important topic.
As far as I know, the derived empirical representer theorem is new. The numerical experiments are somewhat limited and the data sets are relatively small.
The supplementary material contains the proofs.

I would like to propose, that if the manuscript was accepted, then the supplementary material should be published as well.
Author Feedback

Author rebuttal: We thank the reviewers for their comments.

==============
Response to Assigned_Reviewer_4:

Yes, at the end of Eq 6, there is a term missing: k(x,y).


==============
Response to Assigned_Reviewer_5:

Q: sec 4.2: why is alpha the same for all the transformations in the first formula? each transformation can have its own parameter, right? This is not clear in the formula.

A: Yes, each transformation can have its own parameter. We just used alpha as a symbol for parameter, which is not supposed to be tied across transformations. We will change the notation to different symbols in the future version.


==============
Response to Assigned_Reviewer_6:

We thank the reviewer for the comments.